# Plasma Endocan as a Biomarker of Thrombotic Events in COVID-19 Patients

**DOI:** 10.3390/jcm11195560

**Published:** 2022-09-22

**Authors:** Camille Chenevier-Gobeaux, Morgane Ducastel, Jean-François Meritet, Yassine Ballaa, Nicolas Chapuis, Frédéric Pene, Nicolas Carlier, Nicolas Roche, Tali-Anne Szwebel, Benjamin Terrier, Didier Borderie

**Affiliations:** 1Department of Automated Biological Diagnostic, Cochin Hospital, APHP-Centre Université de Paris, CEDEX 14, 75679 Paris, France; 2Department of Virology, Cochin Hospital, APHP-Centre Université de Paris, CEDEX 14, 75679 Paris, France; 3Department of Haematology, Cochin Hospital, APHP-Centre Université de Paris, CEDEX 14, 75679 Paris, France; 4Medical Intensive Care Unit, Cochin Hospital, APHP-Centre Université de Paris, CEDEX 14, 75679 Paris, France; 5UMR 8104, INSERM U1016, CNRS, Université de Paris, 75014 Paris, France; 6Department of Pulmonology, Cochin Hospital, APHP-Centre Université de Paris, CEDEX 14, 75679 Paris, France; 7UMR 1016, Institut Cochin, Université de Paris, 75014 Paris, France; 8Department of Internal Medicine, Cochin Hospital, APHP-Centre Université de Paris, CEDEX 14, 75679 Paris, France; 9Centre de Référence Maladies Auto-Immunes et Maladies Systémiques Rares, d’Ile-de-France, Université de Paris, 75014 Paris, France; 10INSERM U970, Paris–Cardiovascular Research Center (PARCC), Université de Paris, 75015 Paris, France; 11INSERM UMRs 1124, Université de Paris, 75006 Paris, France

**Keywords:** biomarker, thrombotic event, COVID-19, endocan, inflammation, SARS-CoV-2

## Abstract

(1) Background: Endocan is a marker of endothelial dysfunction that may be associated with thrombotic events. The aim of the study was to investigate the performance of endocan as a marker of thrombotic events in COVID-19 patients. (2) Methods: We measured endocan in plasma from 79 documented COVID-19 patients classified according to disease severity (from mild to critical). Thrombotic events were recorded. (3) Results: Endocan concentrations at admission were significantly increased according to COVID-19 severity. Levels of endocan were significantly increased in patients experiencing thrombotic events in comparison with those without (16.2 (5.5–26.7) vs. 1.81 (0.71–10.5) ng/mL, *p* < 0.001). However, endocan concentrations were not different between pulmonary embolism and other thrombotic events. The Receiver Operating Characteristic (ROC) analysis for the identification of thrombotic events showed an area under the ROC curve (AUC) of 0.776 with an optimal threshold at 2.83 ng/mL (93.8% sensitivity and 54.7% specificity). When combining an endocan measurement with D-dimers, the AUC increased to 0.853. When considering both biomarkers, the Kaplan–Meier survival curves showed that the combination of endocan and D-dimers better discriminated patients with thrombotic events than those without. The combination of D-dimers and endocan was independently associated with thrombotic events. (4) Conclusions: Endocan might be a useful and informative biomarker to better identify thrombotic events in COVID-19 patients.

## 1. Introduction

Endocan (endothelial cell-specific molecule 1, or ESM-1) is an endothelial cell-specific proteoglycan, first described in 1996 [1] and expressed by pulmonary endothelial cells [2], although not fully selectively [3]. The structure of endocan includes a polypeptide core of 184 amino acids and a unique chondroitin dermatan sulfate O linked to serine 137 [4]. Endocan is fully secreted as a proteoglycan and does not bind to the endothelial cell membrane or perivascular extracellular matrix. The synthesis and secretion of endocan are upregulated by proinflammatory mediators such as tumor necrosis factor (TNF)-α, interleukin (IL)-1β, bacterial components such as lipopolysaccharide (LPS) or angiogenic factors such as VEGF or FGF-2 [5,6]. Endocan may be involved in molecular interactions with a wide range of biologically active moieties, essential for the regulation of biological processes such as cell adhesion, migration, proliferation and neovascularization [7]. Endocan binds to leukocyte function-associated antigen-1 (LFA-1) and inhibits its interactions with endothelial intercellular adhesion molecule-1 (ICAM-1). Endocan is, thus, able to modulate leukocyte migration from the blood flow into tissues [8,9]. Endocan is released by the pulmonary endothelium in response to local or systemic injury. It mainly inhibits leukocyte diapedesis rather than leukocyte rolling or adhesion to endothelial cells, both in vitro and in vivo [10]. The measurement of endocan in peripheral blood was found to be increased in pneumonia and correlated with disease severity [11]. Güzel et al. also found a correlation between plasma endocan levels and venous thromboembolism [12].

There is growing evidence about the value of endocan in COVID-19 in the literature. The etiopathology of COVID-19-related cardiovascular (CV) disease could involve endothelial dysfunction [13]. Endocan was recently identified as a marker of endothelial dysfunction and potentially as an immune-inflammatory marker that may be associated with CV events [14,15]. There is also some evidence of an association between inflammatory diseases and increased circulating levels of endocan [16]. In this context, endocan is a strong predictor of all-cause and CV mortality in patients with inflammatory conditions [17]. Thromboinflammation, endotheliopathy and coagulopathy are frequent in severe COVID-19 [18], and thrombotic events are associated with worse prognoses in COVID-19 [19]. Very recently, several studies reported on potential interest in serum endocan as a potential marker in the diagnosis and prediction of disease severity in COVID-19 [20,21,22,23,24,25,26], and most of them also mentioned the importance of this marker’s connection with thrombosis. However, none of them specifically focused on the potential role of endocan in identifying thrombotic injury in COVID-19, nor investigate the mechanistic importance of endocan in thromboinflammation.

Thus, the aim of the study was to investigate the performance of endocan as a marker of thrombotic events in COVID-19 patients.

## 2. Materials and Methods

### 2.1. Studied Population

This study was an ancillary study of a previous work [27]. From April 2020 to May 2020, 278 leftover heparinized plasma samples from patients with suspected SARS-CoV-2 infection (COVID-19) were collected in Cochin Hospital. The study was performed according to the principles of the Declaration of Helsinki and was approved by our local ethics committee (institutional reviewing board CLEP N°: AAA-2020-08050). Patients’ clinical and biological data were collected, and included symptoms, comorbidities, routine blood biomarkers (CRP, fibrinogen, D-dimers, leukocyte count, neutrophil count) and COVID-19 status according to RT-PCR results performed at admission and/or seroconversion. Patients were classified according to severity from mild (stage 0) to critical (stage 3), adapted from the NIH COVID-19 Treatment Guidelines [28]: mild cases had symptoms without dyspnea or abnormal imaging (COVID-19 stage 0); moderate cases showed evidence of lower respiratory disease with SpO_2_ > 94% (COVID-19 stage 1); severe cases had SpO_2_ < 94% or respiratory rate >30 or lung infiltrates higher than 50% on computed tomography (COVID-19 stage 2); critical cases presented acute respiratory distress syndrome or septic shock (COVID-19 stage 3). Patients with increased O_2_ needs were defined as requiring oxygen. When clinically suspected, screening for thrombotic events was performed either with an injected CT scan or with medical ultrasound. Thrombotic events were collected (we considered all thrombotic events detected during the screening; the timing of the event was recorded and distant thrombotic events were taken into account) and patients were classified as having pulmonary embolism (PE) or other types of venous thromboembolism (VTE). Finally, endocan could be measured in leftover samples for 79 patients with confirmed COVID-19 (see Appendix A).

### 2.2. Endocan Measurement

Endocan was measured in plasma samples collected at admission (on first blood collection obtained when the patient was admitted to the hospital) using the DIYEK H1 human endocan/ESM-1 ELISA kit, which is based on the immunoenzymatic assay (ref. LIK-1101; Biothelis, Lille, France). The measuring range was from 0.15 to 10 ng/mL. During the study period, the between-assay imprecision was <10%, based on a quality control sample targeted at 5 ng/mL. Endocan is a stable circulating molecule [29]. In healthy subjects, endocan circulates in a range within 0.15 to 2.5 ng/mL.

### 2.3. Statistical Analysis

Continuous variables were presented as median (25th–75th percentile) and categorical variables as numbers and percentages. Continuous variables were compared with the Mann–Whitney *U* test and categorical variables using the Pearson chi-square test. Receiver–operator characteristic (ROC) curves were constructed to assess the sensitivity and specificity and positive (PPV) and negative predictive value (NPV) (all with their 95% confidence interval (95% CI)) throughout the concentrations of biomarkers to compare the accuracy of these biomarkers for the diagnosis of PE. A comparison of areas under the ROC curves was performed. The normality of the distribution was tested with the Kolmogorov–Smirnov test for all investigated biomarkers. When the distribution was not normal, a log-transformation was performed. Log-transformed values were, therefore, used in the subsequent analysis (correlation). The correlation between biomarkers was assessed using the Spearman rank correlation in order to determine the multicollinearity between two variables. We obtained Kaplan–Meyer curves according to biomarker values for thrombotic event occurrence. A *p* value of <0.05 was considered significant. A stepwise logistic regression was performed to assess variables associated with thrombotic events. Biomarkers were log-transformed if the distribution was not normal. Only variables with *p* value < 0.20 in the univariate analysis were included in the regression analysis. The discriminate power of the logistic regression was evaluated with the c-statistic (concordance index) and the goodness of fit of the model with the Hosmer–Lemeshow test. Statistical analysis was performed using MedCalc (MedCalc Software, Mariakerke, Belgium).

## 3. Results

### 3.1. Endocan and Thrombotic Events in COVID-19

The baseline characteristics of the study population according to thrombotic events are presented in Table 1. Patients who experienced thrombotic events (16/79, 20%) were more frequently men requiring oxygen, more frequently admitted to the ICU and had a longer stay in the hospital. These patients also had higher levels of CRP and D-dimers at admission. Levels of endocan at admission were significantly increased in patients experiencing thrombotic events in comparison to those who did not ((16.2 (5.5–26.7) vs. 1.81 (0.71–10.5) ng/mL, *p* < 0.001) (Table 1). However, endocan concentrations were not different between PE and other types of VTE.

The ROC curve analysis of endocan to predict the occurrence of thrombotic events showed an AUC of 0.776 (95%CI: 0.669–0.862; *p* < 0.001), with an optimal threshold at 2.83 ng/mL, 93.8% for sensitivity, 54.7% for specificity, 97.2% for NPV and 34.1% for PPV. In comparison, the ROC curve analysis of D-dimers to predict the occurrence of thrombotic events showed an AUC of 0.793 (95%CI: 0.660–0.892; *p* < 0.001), with an optimal threshold at 2.15 µg/mL, 81.6% for sensitivity, 76.2% for specificity, a 94.1% for NPV and a 47.4% for PPV. There was no significant difference between the two AUCs (*p* = 0.770) (Figure 1). When combining endocan with D-dimers, the AUC increased to 0.853 (95%CI: 0.729–0.935; *p* < 0.001; sensitivity to 81.8%, specificity to 82.0%, NPV to 94% and PPV to 52%) (Figure 1). The combination of endocan and D-dimers better discriminated patients with thrombotic events than those without (Figure 2).

We further performed a logistic regression to assess variables associated with TE. Only variables with *p* values of <0.20 in the univariate analysis were included: gender, overweight/obesity, temperature, COVID-19 severity, CRP, D-dimers and endocan. When included separately, the only variable that was independently associated with thrombotic events was COVID-19 severity (OR 3.9 (95%CI: 1.4–11.4); c-statistic = 0.779; Hosmer–Lemeshow test: Chi^2^ = 0.13; *p* = 0.936). When included in the regression as a combination, D-dimers associated with endocan were found to be the only variable that was independently associated with thrombotic events (OR 2.0 (95%CI: 1.2–3.2); c-statistic = 0.806; Hosmer–Lemeshow test: Chi^2^ = 7.58; *p* = 0.475).

### 3.2. Endocan and COVID-19 Severity

Levels of endocan in peripheral blood at admission were significantly increased according to COVID-19 severity (*p* < 0.0001 for trend) (Figure 3). Endocan levels were also significantly higher in patients with an increased O_2_ need compared to patients without (8.40 (4.25–22.20) vs. 1.00 (0.36–7.25) ng/mL, *p* < 0.001). In contrast, endocan levels were not different in COVID-19 patients according to the presence or absence of arterial hypertension or CV disease. Furthermore, endocan levels were significantly higher in patients admitted to the ICU compared to patients admitted to a conventional ward (14.30 (4.54–25.48) vs. 1.41 (0.68–10.28) ng/mL, *p* < 0.001).

Finally, we analyzed correlations between endocan levels and thromboinflammatory markers, including CRP levels, neutrophil count, neutrophil/lymphocyte ratio, platelets and D-dimer levels. In COVID-19 patients, endocan was significantly but moderately correlated with CRP, neutrophil count, neutrophil/lymphocyte ratio and D-dimer values (Table 2).

## 4. Discussion

In this ancillary study, we aimed to investigate the performance of endocan as a marker of thrombotic injury in COVID-19 patients. Our work reported two major findings: (1) high endocan concentrations were associated with thrombotic events in COVID-19 patients, and (2) endocan concentrations were significantly increased according to COVID-19 severity. We further aimed to generate a discussion on the mechanistic importance of endocan in thromboinflammation.

Our approach was original and focused on the potential role of endocan in identifying thrombotic injury in COVID-19. We reported here that concentrations of endocan at admission were significantly increased in patients that presented thrombotic events. These results were in line with that of Güzel et al., who found a correlation between plasmatic endocan levels and thromboembolism in non-COVID-19 patients [12]. In our study, we found a high proportion of patients with PE among those with a thrombotic event (n = 11 cases out of 16, i.e., 69%) and a minor proportion of DVT (n = 1.6%). These observations may be slightly different from data from the literature. Indeed, overall, studies have suggested that DVT and the thrombotic microangiopathy of pulmonary capillaries are more frequently observed in severe SARS-CoV-2 infection than in PE [19]. Those microangiopathies have been described in several organs, particularly in the lungs, but also kidneys or heart [29,30]. Platelets are abundant in alveolar tissue, and could play a role in the pathophysiology of COVID-19, both in terms of protection through the potential phagocytosis of the virus and aggravation by amplifying local inflammatory processes and increasing the risk of the occlusion of pulmonary capillaries. Of note, in our population we could not identify any correlation between endocan and circulating platelets, and platelets were not significantly decreased in patients with thrombotic events. However, the correlation between endocan and D-dimers was moderate, indicating that both biomarkers had a relative independence. In our study, the combination of endocan with D-dimers appeared to perform as expected to identify thrombotic events, and was independently associated with TE in the multivariate analysis. The pathophysiology of the thromboses described in patients with severe forms of COVID-19 resulted from different mechanisms. SARS-CoV-2 primarily targets pneumocytes, immune system cells and vascular endothelial cells through an interaction between the viral S protein and the angiotensin-converting enzyme 2 (ACE2) receptor. It has been shown that COVID-19-associated coagulopathy—and, in particular, microvascular thrombosis causing alveolar damage—is due to the immune system and to thromboinflammatory responses [31,32]. Alveolar damages and microangiopathies have been demonstrated in patients who died from severe forms of COVID-19 after autopsy and immunohistochemistry techniques on lung biopsies [33,34]. The SARS-CoV-2 spike protein appears to be involved in endothelial damage, leading to the severe forms [35]. The immune system causes the secretion of proinflammatory cytokines such as IL-6 or TNFα, which can travel as far as the cytokine storm and promotes the secretion of procoagulant factors [36]. SARS-CoV-2 can also activate complementary pathways, accentuating endothelial damage and promoting thrombotic events, such as complement-mediated thrombotic microangiopathies [30]. Endocan concentrations may be correlated with the histological evidence of pulmonary microangiopathies in patients according to the degree of the severity of COVID-19; however, this could not be studied in our work. Regardless, we found that combining endocan with D-dimers better discriminated patients with thrombotic events than those without. Furthermore, this combination was found to be independently associated with thrombotic events. This observation allowed us to consider that endocan may have an added value to better identify thrombotic events in COVID-19 patients. The correlation between endocan and D-dimers was moderate, indicating a relative independence from one to the other: if D-dimer production is more likely to be the consequence of a fibrinolysis, endocan levels might be related to the prothrombotic conversion of the endothelial surface. Our results suggested that, in the presence of endothelial dysfunction and endocan secretion into the bloodstream, the likelihood of thrombotic complications in COVID-19 increases. However, endocan cannot be considered as an isolated marker of thrombotic complications without D-dimers. We believe that endocan is not only a marker of endothelial dysfunction. Although some authors believe that soluble endocan can inhibit leukocyte recruitment and diapedesis to endothelium [37], there is strong evidence that endocan is a potential immunoinflammatory marker [38]. Several lines of evidence support the importance of leukocyte-dependent thrombotic events in some pathological conditions, such as DVT, while leukocyte activation and adhesion also play a key role in thrombotic events and COVID-19 [39]. Especially in DVT, the increase in endocan levels as an indicator of endothelial dysfunction cannot overshadow the main mechanism of thrombosis, which is secondary to leukocyte adhesion. On the one hand, soluble endocan is related to the increase in the expression of inflammatory and adhesion molecules, and may have effects on the adhesion of leukocytes and subsequent platelet recruitment to the inflamed endothelium [38]. On the other hand, soluble endocan itself may also have effects on thrombotic functions independent of endothelial dysfunction, such as increasing the level of platelet and leukocyte activity [39].

Secondly, we reported that endocan concentrations significantly increased according to COVID-19 severity. Our results were in accordance with and confirmed previous observations in the literature [11,23,24]. Indeed, blood endocan was found to be increased in pneumonia, correlating with clinical scores of pneumonia severity [11]. Furthermore, elevated blood concentrations of endocan at days 3–4 may distinguish moderate from severe acute respiratory distress syndrome during COVID-19 [23]. Very recently, Gaudet et al. demonstrated that elevated values of endocan were a predictive variable for late acute respiratory failure worsening [24]. In our cohort, plasma endocan levels were moderately correlated with routine inflammatory biomarkers, such as CRP, neutrophil count, neutrophil/lymphocytes ratio and D-dimer values. These results were in accordance with those obtained in previous studies in patients with cardiovascular diseases [16,17] and in COVID-19 patients [20]. We also reported that endocan values at admission were significantly increased in patients admitted to the ICU in comparison with those who were not admitted to the ICU. This finding was in line with the study of Medetalibeyoglu et al., even if endocan was not assayed with the same kit [20]. Very recently, a few studies reported the potential interest of serum endocan in the prognosis of COVID-19 [21,22,23]. These studies focused on endocan’s ability to predict mortality [21] and to distinguish moderate from severe degrees of pneumonia [22,23], with different conclusions. While Guzel et al. reported that endocan was not associated with the degree of pneumonia [22], Pascreau et al. reported that elevated blood concentrations of endocan may distinguish moderate from severe acute respiratory distress syndrome during COVID-19 [23]. Of note, both teams studied endocan at days 3–8 of admission. None of these studies specifically focused on the potential role of endocan in identifying thrombotic injury in COVID-19 [20,21,22,23,24].

Next, we acknowledged the presence of some limitations in this work: First, this was an ancillary study of a previously published work [27] that was designed as a single-center, retrospective study. Second, we could not measure endocan in all patients; thus, the sample size was modest in comparison with previously published cohorts. However, this retrospective study was performed during an epidemic peak, and presented a unique opportunity for the collection of homogenous data from the same outbreak.

## 5. Conclusions

In conclusion, endocan might be a useful and informative biomarker to better identify thrombotic events in COVID-19 patients. Our results highlighted the mechanistic importance of endocan in thromboinflammation.

## Figures and Tables

**Figure 1 jcm-11-05560-f001:**
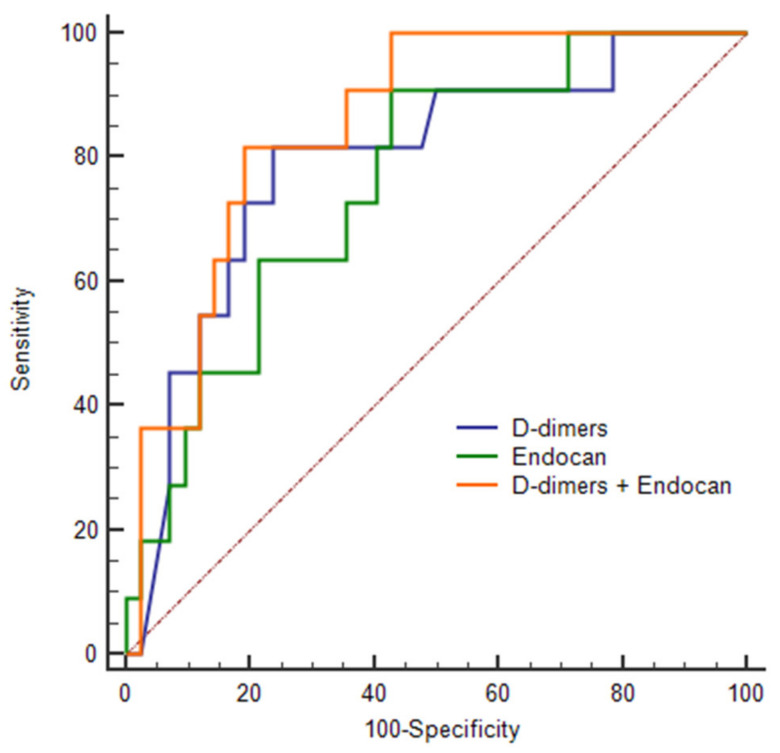
Receiver Operating Characteristic (ROC) curve for D-dimers and endocan to identify thrombotic events.

**Figure 2 jcm-11-05560-f002:**
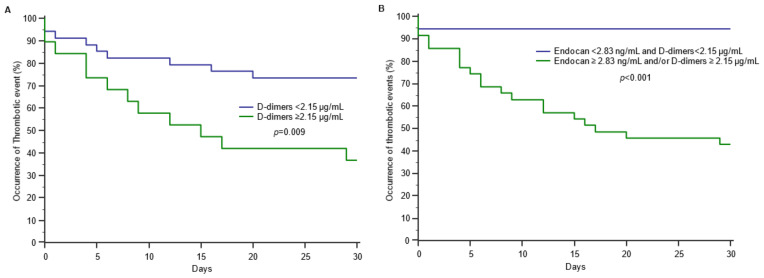
Occurrence of thrombotic events according to D-dimers concentration (**A**) and to the combination of D-dimers and endocan (**B**) in patients with COVID-19.

**Figure 3 jcm-11-05560-f003:**
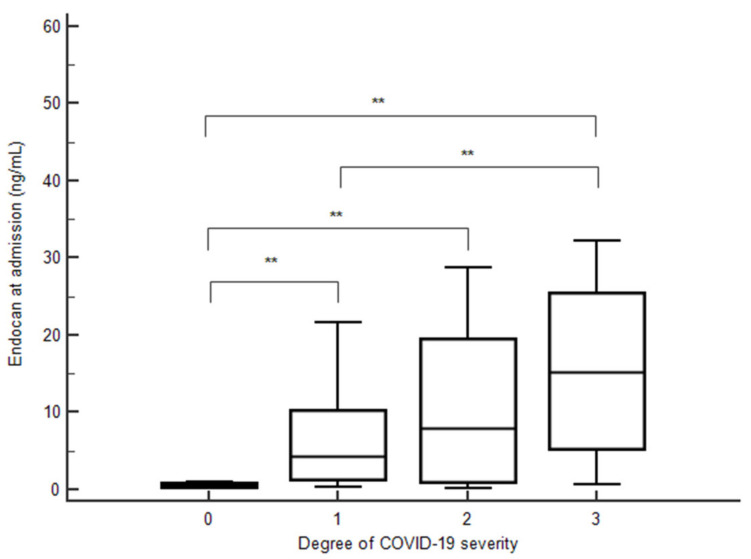
Endocan concentrations at admission according to COVID-19 severity. **, *p* < 0.05.

**Table 1 jcm-11-05560-t001:** Baseline characteristics of the studied population.

Variable	No Thrombotic Event	Thrombotic Event	*p*
**Demographics**			
N	63	16	
Age—years	62 (47–73)	54 (50–66)	0.427
Male—n (%)	31 (49)	13 (81)	0.019
Cardiovascular disease—n (%)	27 (43)	6 (68)	0.700
Overweight/obesity—n (%)	24 (38)	10 (63)	0.080
Hypertension—n (%)	19 (30)	5 (31)	0.841
Diabetes—n (%)	15 (24)	2 (13)	0.631
Chronic respiratory failure—n (%)	3 (5)	0 (0)	0.600
Systemic AI disease—n (%)	2 (3)	0 (0)	0.686
**Symptoms at admission**			
Temperature >38 °C—n (%)	41 (65)	14 (88)	0.084
Dyspnea—n (%)	36 (57)	14 (88)	0.025
Myalgias—n (%)	21 (33)	5 (31)	0.875
Fatigue—n (%)	29 (46)	5 (31)	0.289
Diarrhea—n (%)	19 (30)	3 (19)	0.366
Oxygenation—n (%)	41 (65)	15 (94)	0.030
Admission flow (L/min)	3 (2–4)	5 (2–14)	0.043
Tomodensitometry performed—n (%)	34 (54)	11 (69)	0.283
Extension at TDM—n (%)			
<10%	3 (9)	0 (0)	
10–25%	13 (38)	4 (36)	
25–50%	11 (32)	2 (18)	
50–75%	2 (6)	3 (27)	
>75%	5 (15)	2 (18)	
Increased O_2_ need—n (%)	21 (33)	13 (81)	0.002
Intensive care Unit admission—n (%)	16 (25)	11 (69)	0.001
COVID-19 severity			<0.001
Stage 0	18	0	
Stage 1	18	3	
Stage 2	14	2	
Stage 3	13	11	
Type of thrombotic event—n (%)			
Pulmonary embolism (PE)	0 (0)	11 (69)	
Venous thromboembolism (VTE)	0 (0)	5 (31)	
**Follow-up**			
Length of stay (days)	9 (6–22)	27 (9–49)	0.021
Death—n (%)	1 (2)	3 (19)	0.054
**Blood routine biomarkers at admission**			
CRP—mg/L (IQR)	75 (27–132)	147 (79–246)	0.017
D-dimers—µg/mL (IQR)	1.15 (0.42–2.15)	6.76 (2.83–10.0)	0.003
Neutrophils—G/L (IQR)	4.3 (2.8–7.2)	6.2 (4.4–8.7)	0.133
Neutrophil/lymphocytes ratio (IQR)	4.5 (2.1–8.7)	4.9 (3.2–11.1)	0.531
Platelets—G/L (IQR)	242 (187–356)	237 (169–305)	0.545
Endocan—ng/mL (IQR)	1.81 (0.71–10.5)	16.2 (5.53–26.7)	<0.001

Variables are expressed in numbers (%) or in median (IQR).

**Table 2 jcm-11-05560-t002:** Correlations of admission values in COVID-19 patients.

	CRP	Neutrophils	Neutrophils/Lymphocytes Ratio	D-Dimers	Platelets
**Endocan**	0.468 **	0.390 **	0.347 *	0.345 *	0.093
**CRP**		0.402 **	0.257	0.445 **	0.043
**Neutrophils**			0.725 **	0.517 **	0.307 **
**Neutrophils/Lymphocytes ratio**				0.291 *	0.068
**D-dimers**					0.237

* *p* < 0.05, ** *p* < 0.01. Correlation coefficient absolute values below 0.3 were considered to be weak (green), from 0.3 to 0.7 were moderate (yellow) and above >0.7 were strong (red). Values were log-transformed before correlation.

## Data Availability

The datasets used and/or analyzed during the current study are available from the corresponding author on reasonable request.

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
