# Peer review of "Plasma Endocan as a Biomarker of Thrombotic Events in COVID-19 Patients"

_jcm, 2022, doi:10.3390/jcm11195560_

Round 1
Reviewer 1 Report
General description
This might be an informative study, while it should be noted that several articles have been published regarding the relationship between endocan and COVID-19 and the importance of measuring its serum level with inflammatory and thrombotic conditions. Although the authors tried to provide direct links between endocan levels and some general laboratory markers of inflammation and thrombosis, the novelty of main massage of manuscript is still limited to some extent. However my suggestion to respected authors is to improve the importance and novelty of paper by adding further discussion on the mechanistic importance of endocan in thrombo-inflammation. Therefore, to improve the quality of the article, I suggest the following.
a-So far many studies reported the role of endocan as a potential marker in diagnosis and predicting disease severity in COVID-19 and of course, almost most of them also mentioned the importance of this marker's connection with thrombosis, I think the authors should mention these articles and their abundance, even if the novelty of their own article is overshadowed. Some examples of these research are:
1-Endocan as a potential marker in diagnosis and predicting disease severity in COVID-19 patients: a promising biomarker for patients with false-negative RT-PCR
2-Serum endocan levels on admission are associated with worse clinical outcomes in COVID-19 patients: a pilot study
3-Diagnostic and prognostic value of serum endocan levels in patients With COVID-19
4-Is endocan correlated to ARDS severity or an epiphenomenon of thrombo-embolic disease in COVID?
b- According to the above articles, I think the authors should correctly specify the novelty of their research for the readers. Please correctly underline whether the authors describe endocan only as a marker of endothelial dysfunction or whether a mechanistic value for this molecule in thromboembolism is also of interest in this paper. If so, the involvement of endocan in thrombo-inflammation should be discussed. And of course, during it, it is necessary for the respected authors to correctly and briefly provide updated information about the mechanisms of thrombosis in Covid-19. In this regard, the following articles are recommended:
1- Platelet-leukocyte crosstalk in COVID-19: How might the reciprocal links between thrombotic events and inflammatory state affect treatment strategies and disease prognosis? “In Thrombosis research 2022”
2- Endocan: A Key Player of Cardiovascular Disease: “in Frontiers in Cardiovascular Medicine 2022”
c- Although Gaudet etal JBC 2020) believe that soluble endocan can inhibit leukocyte recruitment and diapedesis to endothelium, this cannot be considered as an anti-inflammatory events. Several line of evidence support the importance of leukocyte dependent-thrombotic event in some pathologic condition such as DVT and DIC while leukocyte activation and adhesion also plays a key role in thrombotic events COVID-19. Especially in DVT, the increase in endocan levels as an indicator of endothelial dysfunction cannot overshadow the main mechanism of thrombosis, which is secondary to leukocyte adhesion. This means that firstly, soluble endocan as an important indicator of endothelial dysfunction is related to the increase in the expression of a range of inflammatory and adhesion molecules, which, independent of LFA-1 and its inhibition, can have profound effects on the adhesion of leukocytes and subsequent platelets recruitment to inflamed endothelium. Secondly, soluble endocan itself may also have effects on thrombotic functions independent of endothelial dysfunction, such as increasing the level of platelet and leukocyte activity. These are important mechanistic points that should be taken into consideration to increase the novelty of the article. As already mentioned in overall the manuscript suffers from limited mechanistic description about different pathways of thrombosis especially in COVID-19.
d- Regardless of discussion and introduction section which need significant improvement, thanks to respected authors in my opinion, designing and conducting experiments and the results obtained from them are generally appropriate and logical.
Author Response
Answer to Reviewer #1: We would like to thank the reviewer for his/her comments. We have revised our manuscript so as to take account of all these comments. We have no rebuttal. All changes are in red script in the manuscript.
- So far many studies reported the role of endocan as a potential marker in diagnosis and predicting disease severity in COVID-19 and of course, almost most of them also mentioned the importance of this marker's connection with thrombosis, I think the authors should mention these articles and their abundance, even if the novelty of their own article is overshadowed. Some examples of these research are:
1- Endocan as a potential marker in diagnosis and predicting disease severity in COVID-19 patients: a promising biomarker for patients with false-negative RT-PCR
2-Serum endocan levels on admission are associated with worse clinical outcomes in COVID-19 patients: a pilot study
3-Diagnostic and prognostic value of serum endocan levels in patients With COVID-19
4-Is endocan correlated to ARDS severity or an epiphenomenon of thrombo-embolic disease in COVID?
As suggested, we reported the role of endocan as a potential marker in diagnosis and prediction in COVID-19 in the introduction of our revised manuscript (lines74-76), and mentioned the abundance of data in the literature citing the additional suggested articles in the revised version of our manuscript (line 74, and ref 25 and 26).
We also improved the discussion on the importance and novelty of our paper by adding further discussion on the mechanistic importance of endocan in thrombo-inflammation (lines 223-224 and 268-285).
- According to the above articles, I think the authors should correctly specify the novelty of their research for the readers. Please correctly underline whether the authors describe endocan only as a marker of endothelial dysfunction or whether a mechanistic value for this molecule in thromboembolism is also of interest in this paper. If so, the involvement of endocan in thrombo-inflammation should be discussed.And of course, during it, it is necessary for the respected authors to correctly and briefly provide updated information about the mechanisms of thrombosis in Covid-19. In this regard, the following articles are recommended:
1- Platelet-leukocyte crosstalk in COVID-19: How might the reciprocal links between thrombotic events and inflammatory state affect treatment strategies and disease prognosis? “In Thrombosis research 2022”
2- Endocan: A Key Player of Cardiovascular Disease: “in Frontiers in Cardiovascular Medicine 2022”
As suggested, we underlined the mechanistic value of endocan in thrombo-inflammation, and we discussed it, providing updated information (lines 223-224 and 268-285). The proposed citations were added to the revised manuscript (ref.38 and 39).
- Although Gaudet et al JBC 2020) believe that soluble endocan can inhibit leukocyte recruitment and diapedesis to endothelium, this cannot be considered as an anti-inflammatory events.Several line of evidence support the importance of leukocyte dependent-thrombotic event in some pathologic condition such as DVT and DIC while leukocyte activation and adhesion also plays a key role in thrombotic events COVID-19. Especially in DVT, the increase in endocan levels as an indicator of endothelial dysfunction cannot overshadow the main mechanism of thrombosis, which is secondary to leukocyte adhesion. This means that firstly, soluble endocan as an important indicator of endothelial dysfunction is related to the increase in the expression of a range of inflammatory and adhesion molecules, which, independent of LFA-1 and its inhibition, can have profound effects on the adhesion of leukocytes and subsequent platelets recruitment to inflamed endothelium. Secondly, soluble endocan itself may also have effects on thrombotic functions independent of endothelial dysfunction, such as increasing the level of platelet and leukocyte activity. These are important mechanistic points that should be taken into consideration to increase the novelty of the article. As already mentioned in overall the manuscript suffers from limited mechanistic description about different pathways of thrombosis especially in COVID-19.
As suggested, we considered the arguments that the reviewer kindly hightlighted for us, in order to increase the novelty of the article (lines 268-285). The cited study of Gaudet et al was added to the reference list (ref 37).
- Regardless of discussion and introduction section which need significant improvement, thanks to respected authors in my opinion, designing and conducting experiments and the results obtained from them are generally appropriate and logical.
We thank reviewer for his opinion.

Reviewer 2 Report
The manuscript is interesting but has a significant bias, so I do not recommend publishing it in its current form. First, the vague term 'at admission' was used. We know that the course of COVID19 changes over time, and it does not matter whether the patient was admitted immediately after the onset of symptoms or seven days later. Second, the patients received different therapies (antiviral, antibacterial, anti-inflammatory) that could significantly change the biomarker profile. It's hard to draw conclusions based on the ROC curve alone, no matter how interesting it looks.
Author Response
Reviewer #2
Comments to the Author
The manuscript is interesting but has a significant bias, so I do not recommend publishing it in its current form. First, the vague term 'at admission' was used. We know that the course of COVID19 changes over time, and it does not matter whether the patient was admitted immediately after the onset of symptoms or seven days later. Second, the patients received different therapies (antiviral, antibacterial, anti-inflammatory) that could significantly change the biomarker profile. It's hard to draw conclusions based on the ROC curve alone, no matter how interesting it looks.
Answer to Reviewer #2: We would like to thank the reviewer for his/her comments. We have revised our manuscript so as to take account of all these comments. We have no rebuttal. All changes are in red script in the manuscript.
- First, the term “at admission” was used in order to indicate that we referred to a single value of endocan on first blood collection obtained when patient was admitted to the hospital. This study was not designed to perform kinetics of endocan. As suggested, we defined more precisely the term “admission” in our revised manuscript, in the method section (lines 108-109).
- Second, we provided a supplemental table describing patients and their therapies. Therapies were given after admission, thus endocan measurement was not affected by them.
- Third, our conclusion was modified and thus are no longer based on the ROC curve alone (lines 316-317).
We hope that these additional informations, together with the revisions proposed by the other reviewers, will bring a new light that may modify the opinion of the reviewer.

Reviewer 3 Report
The study of C. Chenevier-Gobeax et al. is devoted to determination whether the circulating endothelial dysfunction marker endocan can be used to diagnose thrombophilia in COVID-19. The paper examines 79 patients with COVID-19 treated in a hospital for whom D-dimer levels, endocan concentrations, and other laboratory parameters were taken at admission to the hospital. As a result of the work, it is concluded that endocan enhances the diagnostic capabilities of D-dimer in predicting thrombotic events in patients. Although the work is relevant and of unconditional interest to readers of the Journal of Clinical Medicine, the following remarks should be taken into account before it can be published.
Major points.
• It is incorrect to state that endocan is expressed only in pulmanary cells (lines 46-48), as it is considered as a marker of some non-lung cancers (see, for example, https://doi.org/10.3109/10520295.2011.577754)
1. The Introduction should describe in more detail the available data on the role of endocan in COVID-19 (lines 74-75, refs 19-23), as they will allow the reader to get a better idea of ​​what pathophysiological processes this molecule can be a marker of.
2. Although this study is performed on a subgroup of a previously published group of patients, an anonymized list of patients indicating their age, disease severity and comorbidities should be attached to this work. It is also of interest to indicate which therapy the patient received in the hospital, as this may have had an impact on thrombotic and cardiovascular events. In addition, the selection criteria for 79 out of 278 patients are not obvious from the description of the methods.
3. Separately, it should be emphasized that it is not clear from the description of the methods what the authors consider as thrombocytic events, what were the numerical criteria for these events, what timing of the events were considered, whether distant thrombotic events were taken into account, etc.
4. The first conclusion of the authors, indicated in lines 212-213, is not confirmed by the presented results - for this it is necessary to show a plot similar to that shown in Figure 2, but for the endocan.
5. In general, the results of the authors suggest that in the presence of endothelial dysfunction and endocan secretion into the bloodstream, the likelihood of thrombotic complications in COVID-19 increases. However, endocan cannot be considered as an isolated marker of thrombotic complications without D-dimer. This should be emphasized separately in the Discussion.
Author Response
Reviewer #3
Comments to the Author
The study of C. Chenevier-Gobeaux et al. is devoted to determination whether the circulating endothelial dysfunction marker endocan can be used to diagnose thrombophilia in COVID-19. The paper examines 79 patients with COVID-19 treated in a hospital for whom D-dimer levels, endocan concentrations, and other laboratory parameters were taken at admission to the hospital. As a result of the work, it is concluded that endocan enhances the diagnostic capabilities of D-dimer in predicting thrombotic events in patients. Although the work is relevant and of unconditional interest to readers of the Journal of Clinical Medicine, the following remarks should be taken into account before it can be published.
Answer to Reviewer #3: We would like to thank the reviewer for his/her comments. We have revised our manuscript so as to take account of all these comments. We have no rebuttal. All changes are in red script in the manuscript.
Major points.
- It is incorrect to state that endocan is expressed only in pulmonary cells (lines 46-48), as it is considered as a marker of some non-lung cancers (see, for example, https://doi.org/10.3109/10520295.2011.577754)
We modified our revised manuscript according to the reviewer’s remark, and added the cited reference (line 48).
- The Introduction should describe in more detail the available data on the role of endocan in COVID-19 (lines 74-75, refs 19-23), as they will allow the reader to get a better idea of ​​what pathophysiological processes this molecule can be a marker of.
As suggested, we reported the role of endocan as a potential marker in diagnosis and prediction in COVID-19 in the introduction of our revised manuscript (lines 74-76), and mentioned the abundance of data in the literature citing the additional suggested articles in the revised version of our manuscript.
- Although this study is performed on a subgroup of a previously published group of patients, an anonymized list of patients indicating their age, disease severity and comorbidities should be attached to this work. It is also of interest to indicate which therapy the patient received in the hospital, as this may have had an impact on thrombotic and cardiovascular events. In addition, the selection criteria for 79 out of 278 patients are not obvious from the description of the methods.
As suggested, we provided as a supplemental Table with the anonymized list of patients with disease severity, comorbidities and therapies (line 105).
In addition, we described better the selection criteria of the patients in the method section (line 104).
- Separately, it should be emphasized that it is not clear from the description of the methods what the authors consider as thrombocytic events, what were the numerical criteria for these events, what timing of the events were considered, whether distant thrombotic events were taken into account, etc.
As suggested we described better the thrombocytic events in the method section of the revised manuscript (lines 101-102), and also in the supplemental Table.
- The first conclusion of the authors, indicated in lines 212-213, is not confirmed by the presented results - for this it is necessary to show a plot similar to that shown in Figure 2, but for the endocan.
As suggested, we modified the first conclusion in our revised discussion (line 212).
- In general, the results of the authors suggest that in the presence of endothelial dysfunction and endocan secretion into the bloodstream, the likelihood of thrombotic complications in COVID-19 increases. However, endocan cannot be considered as an isolated marker of thrombotic complications without D-dimer. This should be emphasized separately in the Discussion.
As suggested, we emphasized separately the argument of the reviewer in our discussion (lines 268-272).

Round 2
Reviewer 1 Report
Thanks to authors, There is no further comments
Reviewer 2 Report
The revised version could be recommended for publication.